# Coping Strategies and Help-Seeking Behaviors of College Students and Postdoctoral Fellows with Disabilities or Pre-Existing Conditions during COVID-19

Caro Wolfner [1,*], Corilyn Ott [2], Kalani Upshaw [3], Angela Stowe [2,4], Lisa Schwiebert [5,6] and Robin Gaines Lanzi [2]

1 Department of Psychology, The University of Alabama at Birmingham, Birmingham, AL 35294, USA
2 Department of Health Behavior, School of Public Health, The University of Alabama at Birmingham, Birmingham, AL 35294, USA
3 School of Health Professions, The University of Alabama at Birmingham, Birmingham, AL 35294, USA
4 Student Counseling Services, The University of Alabama at Birmingham, Birmingham, AL 35294, USA
5 Department of Cell, Developmental, and Integrative Biology, The University of Alabama at Birmingham, Birmingham, AL 35294, USA
6 The Graduate School, The University of Alabama at Birmingham, Birmingham, AL 35294, USA
* Correspondence: cwolfner@uab.edu

**Abstract:** The COVID-19 pandemic continues to have a global and multifaceted impact on public health. Marginalized and vulnerable populations, such as college students and postdoctoral fellows with disabilities or pre-existing conditions, are being disproportionately affected by the COVID-19 pandemic. Various barriers contribute to an individual's intentions to seek mental health help, but with COVID-19's unprecedented permeability, more research is needed to support this student population. This phenomenological study explored the coping strategies and help-seeking behaviors of college students and postdoctoral fellows with disabilities during the COVID-19 pandemic. Using semi-structured, in-depth interviews from fall 2020 ($n = 36$) and spring 2021 ($n = 28$), a thematic analysis was conducted. The Transactional Model of Stress and Coping was used to triangulate findings, to better understand the relationship between perceived stressors, coping mechanisms, and psychological outcomes. The findings show that (1) college students with disabilities coped in multiple ways (i.e., behavioral, relational, and mental), with some noting improvements in their coping abilities by spring, and (2) stigma prevented college students with disabilities from seeking help when needed. These findings emphasize the need for higher education to address ableism and use factors beneficial to fostering resiliency (i.e., social support, optimism, and self-advocacy) among college students with disabilities.

**Keywords:** coping; help seeking; college students with disabilities; COVID-19; stigma

## 1. Introduction

College students with disabilities or pre-existing conditions, particularly those who also identify with additional marginalized communities, faced compounding and unique challenges as the COVID-19 pandemic began (e.g., making sense of their disability identity, changes in education, lack of social connection, etc.) [1–5]. Some of these challenges were not new, as these individuals had already been experiencing academic and mental health stressors before (e.g., difficulty focusing, overwhelming workload, loss of energy, etc.) [2,6–8]. In comparison to their non-disabled peers, students with disabilities have a higher risk of attrition and lower rates of social and academic adjustment [9]. Due to pandemic-related social distancing measures and on-campus living restrictions, necessary supports and opportunities to socialize and develop relationships with peers

were truncated. It is within these social networks that students often explore their identities [10], and thus, without proper support networks and chances for social engagement, this population is at a compounding risk for social isolation [2,11].

For college students with disabilities or pre-existing conditions, social support from involvement in campus activities and organizations and from faculty is instrumental to feeling a sense of belonging. Importantly, social support can act as a buffer to high levels of perceived stress, minimizing the effects of the stress [12,13]. Additionally, a lack of adequate social support and increased rates of social isolation can lead to poor coping strategies (e.g., substance abuse and unhealthy eating habits) and negatively impact important adulthood outcomes (e.g., academic achievement, independent living, and vocational commitments) [14–16]. This is consistent with current studies illustrating that social isolation and feelings of loneliness have increased alongside the progression of the pandemic. The impact of these emotional difficulties, however, has been disproportionate—affecting young adults more than other age cohorts [17], and more specifically, students with pre-existing mental health and psychological conditions [18].

Given the disproportionate impact on those with disabilities or pre-existing conditions, it is important to understand how these individuals cope or are responding in the face of these stressors and challenges. Additionally, understanding what barriers students with disabilities are facing when support is needed will help us better understand how to maximize university support resources. The Transactional Model of Stress and Coping (TMSC) [19,20] provides a basic framework for explaining the processes of individuals coping with stressful events and thus was used to guide this research. For purposes of conciseness, please note the terms 'college students with disabilities' and 'students with disabilities' are used interchangeably with 'college students and postdoctoral fellows with disabilities and/or pre-existing conditions'.

*Transactional Model of Stress and Coping (TMSC)*

The TMSC can be broken down into five components or tenets: (1) primary appraisal, (2) secondary appraisal, (3) coping efforts, (4) meaning-based coping, and (5) coping outcomes [21]. According to Lazarus and Folkman [20], stress involves an encounter, or transaction, between the self in their external environment. Primary appraisal describes the initial evaluation of a stressor or event, whether individually or by group. At this stage, a stressor is evaluated based on the likelihood the stressor will cause harm or pose a threat (e.g., perception to susceptibility). Following this, an individual will engage in secondary appraisal, where the stressor is evaluated based on the individual's ability to control the stressor and what resources are available to cope with the stressor. An individual's perception of their ability to change the stressful situation at hand, as well as one's perceived ability to regulate associated emotions, are examples of secondary appraisal [22]. An important factor in this appraisal is self-efficacy, or one's beliefs about their capabilities to address certain situations. These appraisals bolster efforts for the individual to engage in coping efforts [19,20].

The two primary coping efforts proposed by the TMSC focus on (1) taking action to minimize or change the stressful situation (problem-focused coping) and (2) redirecting the emotions and thoughts one has toward the stressful situation (emotion-focused coping). Coping outcomes according to Lazarus and Folkman [20] illustrate an individual's successful or unsuccessful ability to adapt to the stressor. Well-being, functional status, and health behaviors are the three primary tenets within this construct [22]. Furthermore, a more recent adaption of this model adds the moderating construct of meaning-based coping, where individuals reappraise or reframe the stressful situation to be perceived in a more positive light (e.g., acceptance, optimism, spiritual practices, etc.). Studies related to coping with chronic stress illustrate that the more exposure one has to a stressor, such as those with chronic conditions or disabilities, the more likely they are to ascribe positive experiences to the situation [21]. Given the heightened stress during emerging adulthood, stress associated with disability (e.g., stigma, self-disclosure, etc.), and the stress of the pandemic, under-

standing how students with disabilities cope is critical [12,23]. Coping research focused on students with disabilities may assist in the development of resilience-building interventions, given their previous experiences with chronic stressors [24]. However, research on the coping experiences of college students with disabilities in the U.S. during the COVID-19 pandemic, specifically from the fall 2020 to spring 2021 academic year, is not present.

The TMSC has been applied across multiple disciplines (such as lifespan, public health, and neuroscience), with more recent applications focusing on understanding health disparities since social and personal factors influence appraisal and coping [22,25]. Additionally, it has been used during natural disasters or times of crisis to assess resiliency overtime [22]. Thus, the incorporation of the TMSC into this study helps in understanding college students' appraisal of stress and perception of coping as the pandemic occurred over the 2020–2021 academic year. This study has broader significance for social institutions to address the inherently ableist society that we live, work, and learn in. The COVID-19 pandemic has been declared a mental health crisis [26], making it imperative to provide inclusive resources and bolster efforts to dismantle the attitudinal and societal barriers that prevent individuals from seeking out resources when needed. Taking steps to reframe how disability is viewed within higher education is essential, eschewing an outdated, ableist, deficit-oriented, and self-stigmatizing view of disability in favor of one that fosters identity development and encourages others to embrace their strengths and values.

By utilizing a phenomenological methodological approach and triangulating the study findings onto the TMSC [20], this research identified coping strategies, the perceived ability to cope with the COVID-19 pandemic, and students with disabilities' support systems. Additionally, findings identified students' perceived barriers from seeking out resources or help when needed. The TMSC [19,20] allowed for a better understanding of the relationships between perceived stressors, coping efforts, and how those constructs influence various psychological outcomes. The TMSC further validated study findings and provided a useful mechanism to illustrate the processes among perceived stressors, the utilization of coping styles, and the psychological impacts students with disabilities experienced during the COVID-19 pandemic. The goal of this study was to explore how students with disabilities and/or pre-existing conditions coped with difficulties encountered during COVID-19. The overarching research question for the study was: what were the self-described coping experiences of college students with disabilities and/or pre-existing conditions as a function of the COVID-19 pandemic during the 2020–2021 academic year? Specifically, how did college students with disabilities cope, and how did they perceive their ability to cope?

## 2. Materials and Methods

### 2.1. Design

For this study's purposes, a phenomenological approach was the best fit, as it seeks to understand a group's lived experiences of a phenomenon, describing the essence of what is being experienced and how it is being experienced [27,28]. The central phenomenon for this study is the self-described coping strategies and help-seeking behaviors of college students and postdoctoral fellows with disabilities and/or pre-existing conditions during the time of the COVID-19 pandemic. When taking a phenomenological approach, data collection methods primarily involve in-depth interviews, as they incorporate open-ended questions to elicit rich information and detailed perspectives on topics which are exploratory in nature [29]. In addition, given the exploratory nature of the study and the complexities of the COVID-19 pandemic, using a phenomenological qualitative research design was warranted. Institutional research ethics (IRB) approval for this project was obtained from the University of Alabama at Birmingham (UAB).

### 2.2. Participants and Recruitment

The parent study for this study is the COVID-19, Race, and Student/Postdoctoral Fellow Mental Health Mixed Methods Study (MPIs: Robin Gaines Lanzi, Angela Stowe, and Lisa Schwiebert), which has three main components: anonymous all-campus sur-

veys of university students and postdoctoral fellows; an 8-month longitudinal study of 122 students and postdoctoral fellows; and stakeholder focus groups. The present study utilized secondary data from the longitudinal phase of the parent study, which included a baseline survey and an initial in-depth interview (fall 2020), follow-up monthly surveys, and a final interview (April 2021). Participants were recruited by a variety of means, such as campus-wide distributed marketing flyers with a QR code. Flyers were also shared across various social media outlets (Facebook, Instagram, Twitter, and LinkedIn), email lists, and UAB student organizations. After scanning the QR code, interested students/postdoctoral fellows were brought to a secure, online survey link via Qualtrics to fill out an interest form which would help determine eligibility. Participants were eligible if they were presently enrolled at UAB as a student or postdoctoral fellow, at least 18 years of age, and willing to consent to the study. After members of the research team contacted the individual and confirmed eligibility, an informational sheet describing the study was given. Once informed consent was obtained, individuals were provided with a secure link to complete the baseline survey. At this same time, participants were assigned a date and time to take part in the initial interview.

For this sub-study, the target population was college students and postdoctoral fellows with disabilities and/or pre-existing conditions. Therefore, from the fall ($n = 122$) and spring interview ($n = 103$) samples, 36 individuals whose disability and/or pre-existing condition was self-identified via fall and/or spring interviews were included. However, 8 participants from the fall interview did not subsequently complete their spring interview and thus could not be included in the spring sample for analysis ($n = 28$).

### 2.3. Interview Procedures

Interviews were conducted with students with disabilities and/or pre-existing conditions during fall 2020 (initial interview, $n = 36$), and again in spring 2021 (final interview, $n = 28$). Specifically, the initial semi-structured in-depth interviews took place from 29 September 2020 to 22 October 2020, while the follow-up interviews took place from 8 April 2021 to 29 April 2021. All the interviews took place on Zoom using a secure passcode and lasted between 25 and 60 minutes. Each participant was assigned a unique participant identification number (PIN) to ensure their privacy. Prior to recording, Zoom usernames were changed to match the PIN. The interviews were conducted by one of seven well-trained interviewers with an IRB-approved interview script to guide both the spring and fall interviews. Upon completion of a recording, Zoom automatically generates audio, video, and transcription files. These files were then uploaded to a password-protected university database. To confirm the accuracy of the transcriptions, audio files were listened to alongside the transcription and corrected as necessary. Given the study's aim, the interview questions chosen for analysis focused on probing how students were coping, how they felt about how they had been coping, who was in their support system, and in times where support was needed, what prevented them from seeking help.

The qualitative analysis of these open-ended questions provided themes and patterns that address our research question about how college students and postdoctoral fellows with disabilities and/or pre-existing conditions are coping.

### 2.4. Thematic Analysis

Using NVivo 12 software (QSR International, Melbourne, VIC, Australia), a secondary thematic analysis of the fall and spring interviews was completed. Thematic analysis is a flexible, reflective, and powerful approach to qualitative analysis, seeking out patterns or meanings within a set of data to derive themes [30–36]. As part of the parent study, two analysts independently coded transcripts and then frequently met to discuss and integrate their coding structures into a single codebook which served as the framework for the fall interviews under the direction of the PI (R.G.L.). Following the completion of the fall interview analyses for the parent study, the final fall codebook served as the initial codebook for the spring interview analyses. In other words, for the spring interviews,

we deductively started with a set of codes (those developed inductively during the fall interviews), and then, as new patterns and themes emerged, we inductively created new codes and iterated on the codes as data analysis continued. It is important to note given the use of secondary data and the incorporation of codebooks, this study is an example of a hybrid approach to thematic analysis, as proposed by Braun and Clarke [35,36].

According to the established codebooks, each transcript was independently coded by the first author (C.W.). Transcripts were additionally coded by the team's head qualitative researcher (C.O.), an expert in qualitative analysis and NVivo, and reviewed for inter-rater reliability. To establish rigor and trustworthiness, they met regularly to discuss areas of matching versus divergent coding and to reconcile differences. The rigor of this process led to a coder agreement of over 90 percent. The process involved deleting, trimming, renaming, and elevating codes to themes. To ensure the themes accurately portrayed, or depicted, the participants' lived experiences, the primary coder (C.W.) conducted iterative reviews, refinements, and creations of codes. According to Braun and Clarke [30], there are five steps to thematic analysis: (1) familiarization of data, (2) initial coding, (3) generating initial themes, (4) reviewing and primary themes, and (5) refining, defining, and naming themes. These steps were taken to conduct the analysis for the broader primary study, as well as the secondary analysis for this study.

This first author familiarized herself with the data to ensure a comprehensive understanding of the depth of the data. This included re-listening to the audio recordings to ensure the transcribed interviews were accurate. Of note, this researcher had extensive involvement in data collection and analysis (i.e., conducting some of the spring interviews and listening to, cleaning, and coding some of the fall and spring interviews). As each transcript was re-read, initial observation notes were taken in a separate document. These observations included statements of potential interest, ideas to explore further, and personal reflections to account for any pre-conceived bias or influences that might have obscured the way the data were read and analyzed. This allowed the researcher to unpack any possible assumptions that may have been underlining initial observations or reactions. Lastly, notes for each data point and/or participant interview were then reviewed to critically evaluate and combine them, reflecting the observations or familiarization with the entire data set.

Once fully immersed in the data, significant statements were annotated alongside the observation notes. Since this study involved secondary analysis, initial codes for the parent dataset were already established. However, thematic analysis is a reflective process, and given the specific interest in students with disabilities, initial codes to capture any discussion of one's disability were created (i.e., Students mentioning disability).

Generating initial themes is an interpretive, active process in which themes are generated from the existing coded data. The initial code of "Students mentioning disability" was expanded to a primary theme of "Pre-existing Conditions and/or Disabilities" by applying sub-themes to represent the distinct experiences of each condition classification (i.e., learning disabilities, psychiatric conditions, mobility/sensory impairments, medical disabilities, etc.). Sub-themes help capture and highlight important tenets of a theme [31,35,36].

Themes already established from the primary analysis of the data were compared to each data item to determine if certain themes were not applicable or if they held enough weight to adequately capture a student's experiences. At this stage, themes could be removed, added, reconceptualized, and combined to ensure validity of each theme [33]. For example, pre-existing sub-themes pertaining to the unsuccessful perception of coping were combined and re-named to capture students with disabilities' coping experiences more coherently. Braun and Clarke [30,35,36] suggest that theme levels or sub-themes should typically stay between 3 and 5. Definitions to certain themes were expanded as well. All adaptions or changes were discussed with the team's primary qualitative coder (C.O.) under the direction of the Principal Investigator (R.G.L.) to ensure continuous validity.

Braun and Clarke's [30] last step in thematic analysis involves refining, defining, and naming themes. At this point, a brief description of each theme was created, and the researcher reviewed each individual theme, identifying the most important aspects to

further illustrate why and how the themes connect with the study's research questions. The final themes and sub-themes were written to create a cohesive narrative that best exemplifies the phenomenon being studied, which is the lived experiences of students with disabilities during the COVID-19 pandemic.

Reliability and trustworthiness were established in a variety of ways: inter-coder agreement, the use of high-quality recording devices, transcribing interviews via a computer-assisted device, listening back to audio files to ensure transcription was identical (cleaned), using a common platform to conduct data analysis, comparing coding across multiple coders, and refining and finalizing the codebook for coding [32,37,38]. To illustrate the validity and strength of this qualitative study, numerous strategies proposed by Creswell and Poth [37] were utilized. These included: (1) clarifying researcher bias and engagement in reflexivity (bracketing out experiences and the use of a research journal), (2) generating a thick and rich description for each theme, and (3) having a professional external source review or peer debrief the data and research process. This allowed the researcher to have a mediator asking questions about the phenomenon and how it was constructed and interpreted [39]. Furthermore, peer debriefing sessions can provide support and reassurance by actively listening to the researcher's vocal expression of emotions [37]. On multiple occasions, the first author (C.W.) met with the research team (two analysts and the PI, R.G.L.) to ensure the results and methodology were well understood and connected.

## 3. Results

### 3.1. Demographics

Out of the 36 participants, the sample was predominately (75.00%) female and White (55.56%). The average age of participants was 25.64 years old (SD = 7.67). Pertaining to grade classification, the participants included 17 graduate students (44.27%), 14 undergraduates (38.9%), and 5 postdoctoral fellows (13.89%). Additional demographic characteristics are presented in Table 1.

**Table 1.** Sociodemographic characteristics of participants.

| Baseline Characteristic | Students with Disabilities or Pre-Existing Conditions *n* = 36 | |
| --- | --- | --- |
| | *n* | % |
| Disability Classification | | |
| Attention-Deficit/Hyperactivity Disorder (ADHD) | 7 | 19.44 |
| Mobility/Sensory Impairment or Medical Disability | 6 | 16.67 |
| Psychiatric Disorder | 26 | 72.22 |
| Traumatic Brain Injury (TBI) | 1 | 2.78 |
| University Designation | | |
| Graduate | 17 | 47.22 |
| Postdoctoral Fellow | 5 | 13.89 |
| Undergraduate | 14 | 38.89 |
| Gender | | |
| Cisgender Female | 27 | 75.00 |
| Cisgender Male | 6 | 16.67 |
| Gender-Nonconforming | 1 | 2.78 |
| Non-Binary | 1 | 2.78 |
| Other | 1 | 2.78 |
| Age | | |
| Less than 19 | 5 | 13.89 |
| 19–20 | 5 | 13.89 |
| 21–25 | 12 | 33.34 |
| 26–30 | 6 | 16.68 |
| 31–35 | 5 | 13.89 |
| Over 35 | 4 | 11.12 |

**Table 1.** *Cont.*

| Baseline Characteristic | Students with Disabilities or Pre-Existing Conditions $n = 36$ | |
|---|---|---|
| | *n* | % |
| **Ethnicity** | | |
| Hispanic or Latino | 3 | 8.33 |
| Non-Hispanic or Latino | 33 | 91.67 |
| **Racial Identity** | | |
| American Indian or Alaskan Native | 0 | 0.00 |
| Asian | 2 | 5.56 |
| Black or African American | 8 | 22.22 |
| Native Hawaiian or Other Pacific Islander | 0 | 0.00 |
| Other Race | 2 | 5.56 |
| Two or more races | 4 | 11.11 |
| African American and Filipino | 1 | 2.78 |
| American Indian and White | 1 | 2.78 |
| Asian and White | 2 | 5.56 |
| White | 20 | 55.56 |
| **US Citizenship** | | |
| US Citizen | 30 | 83.33 |
| Non-US Citizen | 6 | 16.67 |
| Permanent resident | 2 | 5.56 |
| International | 3 | 8.33 |
| Non-US National | 1 | 2.78 |
| **School** | | |
| Arts and Sciences | 14 | 38.89 |
| Business | 2 | 5.56 |
| Dentistry, Medicine, Optometry | 4 | 11.11 |
| Education | 1 | 2.78 |
| Engineering | 2 | 5.56 |
| Graduate | 4 | 11.11 |
| Health Professions | 1 | 2.78 |
| Joint Health Sciences | 4 | 11.11 |
| Nursing | 1 | 2.78 |
| Public Health | 3 | 8.33 |
| **Projected Graduation Year** | | |
| 2020 | 3 | 8.33 |
| 2021 | 9 | 25.00 |
| 2022 | 9 | 25.00 |
| 2023 | 4 | 11.11 |
| 2024 | 9 | 25.71 |
| 2025 | 1 | 2.78 |
| 2026 or later | 1 | 2.78 |

The Disability Support Services (DSS) guidelines at UAB were used to classify each student's self-identified disability or condition. Accordingly, 77.22% of the sample were identified as having a psychiatric condition ($n = 26$), 19 % of the sample had ADHD ($n = 7$), followed by 16.67% of participants having a mobility/sensory impairment or medical disability ($n = 6$). Specific psychiatric conditions included depression ($n = 6$), anxiety ($n = 3$), anxiety and depression combined ($n = 9$), obsessive compulsive disorder (OCD; $n = 1$), and an eating disorder ($n = 1$), and two participants had bipolar disorder in addition to their comorbid anxiety and depression (Table 2).

**Table 2.** Sample breakdown of disabilities or conditions.

| Disability or Condition(s) | Students with Disabilities or Pre-Existing Conditions $n = 36$ | |
| --- | --- | --- |
| | *n* | % |
| ADHD | 3 | 8.33 |
| ADHD and Anxiety | 1 | 2.78 |
| Anxiety | 4 | 11.11 |
| Anxiety and Depression | 9 | 25.00 |
| Anxiety, Depression, and Bipolar | 1 | 2.78 |
| Asthma | 4 | 11.11 |
| Depression | 6 | 16.67 |
| Depression and ADHD | 2 | 5.56 |
| Depression and Bipolar | 1 | 2.78 |
| Depression, Eating Disorder, and ADHD | 1 | 2.78 |
| Epilepsy | 1 | 2.78 |
| Muscular sclerosis (MS) and Asthma | 1 | 2.78 |
| OCD | 1 | 2.78 |
| TBI | 1 | 2.78 |

*3.2. Qualitative Findings*

In response to the coping-specific questions, three themes were identified which represent the main groupings of coping strategies. These included mental, behavioral, and relational coping strategies. For this study, these strategies are not identified as positive or negative, because a positive coping strategy for one student could be a negative strategy for another. However, coping strategies were deemed by the students as being unsatisfactory or satisfactory. The analysis identified four sub-themes to express why they believed their coping strategies were unsatisfactory. On the contrary, the analysis also revealed four sub-themes which identified why students with disabilities felt their coping was satisfactory: managing stress, less isolating, productive, and accepting changes. During the spring analysis, an additional theme of "Improving Over Time: Resilience and Perseverance" emerged as more students with disabilities felt their coping strategies and their ability to cope had improved since their initial fall interview.

Lastly, students with disabilities were asked to describe who makes up their support system and to illustrate a time in which help was needed but not sought out and why. The analysis of these support-based questions revealed two primary themes (Support System and Perceived Barriers to Help Seeking) with three sub themes within the first primary and three sub themes for the last primary theme of 'Perceived Barriers to Help Seeking'.

Additionally, by using cross-tabulation queries in NVivo 12, we were able to cross-tabulate codes or themes against attributes (i.e., demographic characteristics). In other words, cross-tabulation analysis was utilized to explore whether unique patterns or experiences with emergent themes were drastically different or similar within demographic groups, specifically across racial identity groups and grade classification.

3.2.1. Theme 1: "Behavioral Coping Strategies"

Behavioral coping strategies included activities in which students with disabilities engaged in or changed their behavior to cope. Many students with disabilities partook in a variety of behavioral-based coping activities, and these were broken down as follows: hobbies, outside activities, routine, over/under/stress eating, and decreased media exposure.

*Hobbies*. Most students with disabilities engaged in a variety of hobbies including reading, watching movies and TV, playing video games, listening and playing music, writing, exercising, and spending time on social media. One female graduate student with OCD noted how her emotional support animal (ESA) helped her cope as she engaged in multiple hobbies at the time of her fall interview:

*"More than anything, just to keep a good headspace I exercise, and I also have an ESA, that's definitely very helpful when it comes to coping, but mostly exercise and then the rest of the time is spent studying."*

In the spring, students with disabilities continued to engage in many hobbies. This included yoga, reading books, playing video games, cooking, journaling, and gardening. A female postdoctoral fellow with asthma expressed how listening to music and joining a discord helped her relax:

*"Music really helped a lot lately ( . . . ) produces a very relaxing effect. So that has been more helpful than I was anticipating ( . . . ) I have also joined a discord or chat board for one of the video game streamers that I like to watch and it's been a good place to decompress."*

*Physical Outdoor Activities.* Many students with disabilities coped by engaging in a variety of forms of physical activities including exercise, walking, sports, and more. Some students with disabilities discussed taking walks, going on runs, and finding relaxation from enjoying nature.

A female postdoctoral fellow who received a diagnosis of anxiety following the onset of the COVID-19 pandemic expressed how being outside was a better coping mechanism for her anxiety compared to pharmaceutical medication. She voiced to the interviewer in her fall interview:

*"The medication is not as helpful, ( . . . ) I feel like a walk it's more appropriate for me because I'm able to enjoy nature and enjoy the weather outside. Enjoy just another environment other than this apartment, so I feel like that's the best thing."*

In the spring, a female graduate student with asthma mentioned hiking on the weekends to cope, *"On the weekends I started hiking and working out more. So those are ways that I feel I've been coping with what's been going on."*

*Routine and Structure.* Many students with disabilities also coped by keeping organized, compartmentalizing, and creating a routine or structure to their day. A female graduate with anxiety and depression explained how creating a routine was anxiety-reducing:

*"Making a schedule was really helping me, especially at times when we were having exams close by, and my schedule becomes really packed with things that have to get done. When I have a schedule made, a planned routine, it really helps me decrease my levels of stress."*

A female undergraduate student with anxiety and depression talked about the difficulty of maintaining a routine and her mental health in her fall interview when she stated, *"Routine does tend to be helpful for me, but my mental health has really kind of thrown some gears in here and there, where I just face some internal exhaustion that makes it too difficult here and there."* A female graduate student with ADHD and anxiety voiced in her spring interview the multiple coping strategies she engaged in, for example, having a set schedule, as she stated: *"Eating healthy and exercising more and setting a schedule has been really helpful to cope"*.

*Decreased Media Exposure.* The circumstances of the pandemic in conjunction with the events related to racism and racial injustice negatively impacted the mental health of many students with disabilities. These events played out on the news and across social media outlets (i.e., Twitter and Instagram) which was overwhelming for some students with disabilities. To cope, many students with disabilities spent less time watching the news and engaging on other social media platforms. A female undergraduate student with ADHD, ED, and depression noted during her fall interview, *"Taking time away from social media has been helpful. I spent the past month off of Instagram and that was really great"*.

At the time of her spring interview, a female graduate student with MS and asthma felt paying less attention to the online media was one way she has learned to adapt to benefit her mental health during the pandemic:

*"Staying off news and social media, that's really helped. Over the last year, not paying attention to the news is great, but you want to look to where to find stuff more positive*

*( . . . ) so there there's a lot of things I've learned, how to almost change or adapt to what I needed for my mental health".*

*Overeating, Undereating, and Stress Eating.* Some students with disabilities engaged in unhealthy eating habits to cope, whether it was overeating, undereating, or stress eating. A female undergraduate student with anxiety, depression and bipolar disorder struggled with maintaining healthy eating habits at the time of her fall interview, when she said, *"My eating habits have been haywire. There will be days where I won't eat, days where I eat too much."* A female graduate student with depression mentioned how this coping mechanism was influenced by academic stress by saying in her fall interview, *"With the increased college responsibilities it makes me more anxious, so I usually turn to food."* In the spring, some students with disabilities were still engaging in these poor eating habits. A demi-guy graduate student with anxiety and depression illustrated this in the spring interview, *"My stress eating has gotten a lot worse. Cookies every night and ice creams. Not ideal".*

3.2.2. Theme 2: "Mental Coping Strategies"

Mental coping strategies included those used to change the student's frame of mind, either positively or negatively. Five sub themes were identified to represent these strategies; these included counseling, distractions, positive thinking (optimism), religious practices or spirituality, and withdrawal.

*Counseling.* The mental coping strategy utilized most by students with disabilities was seeking out professional help, such as counselors, therapists, and other support services. Some students with disabilities sought out these services for the first time during COVID. A female postdoctoral fellow with anxiety and depression discussed the help her counselor had provided when stating in her fall interview, *"I started seeing a counselor to help with that and she's giving me some tips on managing my anxiety."* In his spring interview, a male postdoctoral fellow with ADHD mentioned seeing a psychiatrist and counselor but, compared to the beginning of the pandemic, was seeing his counselor less. He stated, *"I see a psychiatrist regularly, oddly enough, I was going in counseling a lot more often towards the beginning of the pandemic than I am now. I haven't had a counseling appointment and you know, maybe 4ish months."*

*Distraction.* Some students with disabilities also coped by engaging in distraction techniques or activities to stay busy or find ways to occupy their mind. Most times, students with disabilities felt distractions were beneficial to their mental health. In the fall, a female undergraduate student with anxiety and depression illustrated how keeping busy gave her purpose:

*"I like to stay busy. That helps with a lot of my anxiety and depression. If I'm staying busy, then I feel like I'm doing something and I feel I have a purpose almost as silly as that sounds, not thinking about it really."*

Some students with disabilities discussed how by engaging in other activities, they were able to distract themselves from thinking about the stressful events going on (i.e., COVID-19, racial injustices, etc.). As illustrated in the spring by a female undergraduate student with depression:

*"I've been trying to stay busy with school. ( . . . ) If I think about it too much I'll get sad about it. I found ways of doing things to take my mind away from it."*

*Positive Thinking (Optimism).* By changing their frame of mind, some students with disabilities used positive thinking and tried to look on the bright side to cope. By taking time to reflect on their experiences, these students with disabilities were able to combat negative thoughts or emotions they had been facing. Although they were struggling with various stressors, reframing their mindset allowed them to process their emotions, see silver linings, and recognize opportunities for growth. A female graduate student with ADHD voiced to the interviewer in the fall about how taking time to reflect reduced her anxiety:

*"I took a course on critical thinking and that taught me mindfulness and self-reflection. So I was trying to put time towards myself to self-reflect, understand my feelings, try to understand why I'm feeling like this, and to try to find ways to cope with it. Sitting down with myself and self-reflecting was really helping me to kind of manage this level of stress that I was under."*

Additionally in the fall, a female graduate student with depression illustrated optimistic thinking by noting that although the pandemic has been difficult, it has been helpful for her to celebrate the small steps or victories along the way:

*"You have to count the victories whatever they are no matter how you got to that point, you're still alive. You're still thriving in some shape form or fashion you haven't dropped out of school. You haven't quit your job. You pay your bills most of the time, so I would say I'm doing pretty great in that. I like to try to be optimist about things."*

In the spring, more students with disabilities discussed practicing mental-based coping in the form of positive thinking than they shared in the fall interviews. They discussed making efforts to reframe negative thoughts and emphasized the importance of being kind to oneself. A male postdoctoral fellow with anxiety expressed the benefits he felt by adopting a positive outlook on the pandemic when he said in his spring interview:

*"I think with time it was more about accepting that this is here to stay for a while, so to not feel bad all the time for things not being perfect, having a positive outlook has helped."*

*Religious or Spirituality Practices.* A few students with disabilities coped by practicing various forms of religion or spiritual practices such as meditation and praying. A female graduate student with anxiety and ADHD illustrated this in her fall interview when she expressed the encouragement she receives from her Bible Study, stating:

*"Watching church online is helpful and I have several friends from a Bible study that we all encourage each other."*

In the spring, mediative and other spiritual practices were continuously utilized as a coping strategy for some students. A female graduate student with a traumatic brain injury felt more peaceful after engaging in spiritual practices when she said in her spring interview: *"Saying prayers, I'm Muslim, sometimes we have some prayers and readings from the books. This also helps me feel more relaxed and peaceful"*.

*Withdrawal.* Some students with disabilities also coped by withdrawing, coming to a place that helped them "escape reality" or disengage from thinking about any stressors. This was done through watching movies, television, reading books, and computer games. While some felt withdrawing or isolating themselves was a helpful coping strategy, others did not. For example, a male graduate student with depression felt withdrawing was a negative coping mechanism for him when he expressed in his fall interview:

*"It's a form of escapism, a lot of the things I watch or do are a form of that. I'm just trying to go to another place and it's also not helping me because I'm just avoiding the issue entirely and there's not much I can do about it. So I still avoid it."*

In the spring, some students with disabilities continued to engage in activities to help them disconnect from the stressors they had been feeling. Others were overwhelmed with work and school and would isolate themselves in response. For example, a male graduate student with depression expressed in his spring interview that due to the workload of his dissertation, he isolated himself from others. He stated, *"More recently, I've been socially isolating myself from my support network because I've been pretty busy with other stressors."*

### 3.2.3. Theme 3: "Relational Coping Strategies: Remaining Connected with Others"

Staying connected with others, whether virtually or in person, provided a lot of support for most students with disabilities. Relational coping helped them feel understood, as their peers could relate to the experiences they were going through. Staying in touch with others was the most utilized coping mechanism across all coping styles (i.e., mental

and behavioral) among the students with disabilities in both the fall and spring interviews. Many students with disabilities discussed that even though some interactions were online, connecting with others helped them be more social and feel less isolated.

In the fall, a female graduate student with asthma exemplified this when she stated: *"Trying to have conversations about how I'm feeling and then for the social parts, even if it has to be over zoom, just joining clubs and stuff like that so I still can talk to other people."* In times of stress, students with disabilities sought support from friends, loved ones, and family members. A female undergraduate student with depression and anxiety expressed how COVID-19 increased the importance of having social support for her:

> *"I've been trying to be really conscious of that and make time to see people and socialize, because that is really important to me and knowing that finding your people that you can talk to about life and stressors and having those people in your life that you can lean on a little bit with everything being really difficult right now, I feel like it's really important."*

Similarly, a male graduate student with ADHD spoke about increasing his attempts to socialize since COVID-19. He expressed in his fall interview: *"I'm talking to friends more as much as I can, reaching out to people that I have maybe I haven't spoken to in a while, I feel like that's helped."*

### 3.2.4. Theme 4: "Satisfactory Coping Perceived"

Students with disabilities were asked to describe how they believed they had been coping with the pandemic. Students with disabilities identified four sub-themes to express why they believed their coping strategies were satisfactory. The following indicators of successful coping are listed in the order from most endorsed to least. These included: (1) managing stress, (2) accepting changes, (3) less isolating, and (4) encouraging productivity.

*Managing Stress.* Most students with disabilities felt that at some point, they were able to be successful when it came to managing their stress. By using their coping mechanisms, students with disabilities felt they were able to reduce some of the anxiety or stress they had been feeling. Students with disabilities voiced these perceptions regarding their overall coping in addition to their perceptions on coping with a specific event.

A female graduate student with OCD illustrated to the interviewer in the fall how exercise has helped release her stress. She stated, *"Physical movement definitely helps clear my head when I'm stuck in a chair all day doing work and stuff and finally being able to release any stress. That's definitely super super helpful."* Some students with disabilities felt proud in their efforts to cope and voiced how prioritizing their mental health helped reduce or manage stressors better. A female undergraduate student with depression demonstrated this in her fall interview when stating, *"I've been recently working on it a lot and trying to make my mental health and everything better. I'm proud of myself for that."* A female undergraduate student with asthma further exemplified this in her fall interview when talking about her decision to address her unhealthy eating habits. She stated:

> *"I'm proud of myself for booking an appointment with my nutritionist and . . . I feel like that's a positive step in like this stress eating part, we like worked out like an eating plan. Maybe I can start eating a little bit better. I'm taking control in that aspect."*

Like in the fall, in the spring, almost all students with disabilities felt their coping helped manage their stressors. A non-binary postdoctoral fellow with anxiety and depression felt their stress was reduced following an uptake in their medication dosage; they stated in the spring, *"I started to take a medication, antidepressant and increased my doses during this time, following my doctor, and I think it was good because this decreased my anxiety and I'm coping better after that".*

*Accepting Changes.* Some students with disabilities felt that their coping efforts were successful because of their ability to accept the changes and disruptions brought about from the COVID-19 pandemic. They were able to learn from the situation, adapt accordingly, and move forward with their life. A male graduate student with depression and bipolar disorder

saw the pandemic as an opportunity to grow by overcoming its associated challenges. He pointed out in his fall interview:

> *"At times you get these thoughts but I just would say it's a phase and it's going to go away and we need to just accept it because in life there will be many challenges and with each challenge or each phase we tend to get better."*

A non-binary postdoctoral student with anxiety and depression discussed their ability to learn new things each day and accept what comes. They indicated in their fall interview, *"I'm feeling that I'm learning. Day by day. Little things. It's making me laugh and more making me happy to see that."* In the spring, more students with disabilities were able to 'see the light at the end of the tunnel', noting feeling better by accepting the situation at hand. A female graduate student with MS and asthma illustrated this when saying:

> *"I'm trying to take life as it comes COVID-19 changed a lot but life is what it is, and through journaling, trying to remind myself, I would write about it. I should really live and let go, things will happen as they happen, and so that's really made me more mindful and also lessened Imposter Syndrome."*

*Less Isolating.* Some students with disabilities said that their coping strategies helped them to feel less isolated, indicating they were content with their ability to cope. Keeping in contact both virtually and socially distanced, whether it was with friends, family, or peers, reduced the isolation that students with disabilities were facing. A female undergraduate student with anxiety and depression felt being around her roommates made things less isolating. In the fall, she expressed to the interviewer, *"When we first went into quarantine it really made me feel isolated and since we're not on like lockdown anymore, I don't feel as isolated, especially since I moved in with some friends."* In the spring, a female postdoctoral fellow with depression and ADHD noted going outside was helpful in combating the stress and isolation she had been feeling:

> *"Going outside and walking and being outdoors, especially with the warm weather kind of helps me not feel as trapped and isolated, so that's been really nice getting out and seeing other people too has benefits from a distance."*

*Encouraging Productivity.* A few students with disabilities also felt their coping was successful due to their ability to stay productive given the additional difficulties of COVID-19. Whether it was taking time for oneself, implementing a routine, or taking time to connect with others, these activities provided some students with disabilities clarity and structure to continue with their responsibilities. As a male graduate student with ADHD mentioned, exercise helped clear his head and manage his thoughts better, making him feel more productive. He stated in the fall, *"Working out does make me feel more productive, I feel more clear. I'm able deal with some of the thoughts whirling around in my head."* Those who felt their coping led to productivity typically engaged in establishing a routine or schedule to their day. For example, a female undergraduate student with anxiety and depression mentioned creating a schedule promoted productivity and enforced a more positive self-view. She expressed in her fall interview, *"It's helping because I can tell on days where I stick to a schedule, I feel better about myself and I feel I've gotten more done and I feel more productive"*.

### 3.2.5. Theme 5: "Unsatisfactory Coping Perceived"

Although most students with disabilities believed they engaged in healthy coping strategies, the thematic analysis revealed four sub-themes identifying unsatisfactory coping strategies. These included: (1) difficulty sustaining self-perceived adequacy and regularity, (2) unable to regulate negative emotions, (3) engaging in negative habits, and (4) procrastination and lack of motivation.

*Difficulty Sustaining Self-Perceived Adequacy and Regularity.* Of the students who were dissatisfied with their coping strategies, most mentioned their inability to engage in coping strategies consistently. Students with disabilities felt certain coping strategies were too dif-

ficult to maintain, despite being able to sustain them previously. During the fall interviews, a male postdoctoral fellow with anxiety felt he was not coping well and was not aware of how to cope even though he sought support from a counselor:

> *"I am not coping very well when it comes to the personal side of things and I'm working with my therapist to deal with that part because I'm still not coping very well. I don't know what to do to cope."*

Many students with disabilities felt they were not able to cope successfully, leaving many upset or disappointed. Some wished they were more productive and had better time management skills, as they struggled to separate work, school, home, and social time. A female undergraduate student with anxiety and depression voiced in her fall interview that she felt her level of productivity and organization had decreased since COVID-19, while the stress she experienced increased:

> *"I tend to be very harsh on myself. I try to hold myself to a really high standard and just feel like I haven't been able to maintain the level of organization and productivity that I usually do during this time. Just because it has put so much stress, which has been very detrimental to my mental health."*

In the spring, many students with disabilities continued to struggle with maintaining and managing their coping, expressing a desire to better balance the key areas in their life (e.g., school, work, self-care, etc.). Many students with disabilities were self-aware of the consequences of not keeping up with their coping, such as not having the time to focus on self-care. A female undergraduate student with anxiety and depression illustrated: *"I can see how COVID-19 is affecting my health and how my coping strategies are falling off. when I fall off that rhythm, I start neglecting self-help items like taking a shower."* A female graduate student with a TBI expressed struggles managing her time to separate work, school, home, and life. She stated, *"I'll skip assignments not do them because I'm feeling very overwhelmed. I couldn't balance coping with my mental health issues and homework"*.

*Unable to Regulate Negative Emotions.* Decreasing or regulating negative emotions they were experiencing was another area that students were dissatisfied. They felt depressed, anxious, guilty, sad, and angry, which impeded their ability to successfully cope. Given most of these students have conditions where such negative emotions are experienced at a clinical level, having them exacerbated by the pandemic made it difficult to maintain healthy coping practices. In the fall, a female undergraduate student with epilepsy highlighted her difficulties coping on days that were more emotionally challenging: *"Being inside all the time, kind of dampens the mood and I try to like cope with humor, but sometimes that doesn't work and sometimes I just have really, really, really bad days."* At the time of her fall interview, a female undergraduate student with asthma felt guilty and unsatisfied with her ability to cope when saying, *"I don't think I'm coping healthy at all but I take naps during the day and that also makes me feel guilty because I should have been working but I'm also tired because I didn't sleep"*.

*Engaging in Negative Habits.* Some students with disabilities also reverted to or overindulged in habits that were not positive or productive. They were aware that these coping mechanisms were not healthy and felt unsatisfied with their coping efforts. These students with disabilities engaged in unhealthy eating habits (e.g., over, under, stress eating), substance use (e.g., nicotine, alcohol, and marijuana), and excessive time on social media platforms (e.g., Instagram and Tik Tok). At times, a few students with disabilities engaged in more than one of these habits.

One example came for a female undergraduate student with asthma who said:

> *"Stress eating—It got really bad. I'd buy a family sized bag of chips and I just eat it in one sitting and you feel terrible after you eat that. There's some other coping things, I'd go on Tik Tok for 5 hours at a time, so you feel guilty after that too. I don't think I'm coping healthy at all."*

Although engagement in negative habits continued into the spring, some students with disabilities mentioned that these negative habits had decreased since the start of the

pandemic. Additionally, many were aware of the issues and had begun taking purposeful action to address them. A female graduate student with depression felt her negative eating habits were better managed at the time of the spring interviews, declaring, *"I have very unhealthy eating behaviors, I do a lot of stress and emotional eating. So yeah, that's not helpful. I think I kind of managed it"*.

*Procrastination and Lack of Motivation*. Coping strategies were perceived as unsuccessful by a few students with disabilities because they struggled to stay motivated, which hindered their progress in completing their daily responsibilities. They also expressed how their lack of productivity or difficulty staying motivated impacted how they viewed themselves. In the fall, a female undergraduate student with depression illustrated this when she stated:

> *"I feel I'm not as productive and that makes me feel really inadequate sometimes . . . because if I sit here on my phone for like 3 hours and I feel really guilty about that and it makes me feel not good about myself. Also, I don't have the motivation to actually do work, so it's this really bad balance."*

Another student, a male graduate student with ADHD, expressed his challenges with separating his productivity and personal self-worth, saying:

> *"I have ADHD and I'm on medication for it, so I've kind of struggled with productivity at times, but this has been a new lesson in that struggle. It is impossible to dissociate my feelings of self-worth from my productivity."*

In the spring interviews, students with disabilities looked back at their coping overall and felt they should have done more in terms of coping. As a demi-guy graduate student with depression and anxiety expressed: *"I wish I'd been more productive and found a way to be useful to people instead of just focusing everything on finishing my degree"*.

### 3.2.6. Theme 6. "Improving over Time: Resilience and Perseverance"

In the final spring interview, we wanted to gain more clarity on how coping had evolved or adapted since the initial fall interview. Unlike the fall interview, the question stem *"how have your coping strategies evolved over the last year?"* was added as a part of the spring interview coping questions. As such, this theme was only present for the spring analysis. Many students with disabilities primarily felt their coping skills and overall management of the COVID-19 pandemic had improved over time. Although difficult, they discussed how they learned and adapted; prioritized their mental health more; reorganized their approaches to balancing work, school, and home; and began accepting the 'new normal'. Over time, students with disabilities continued to implement helpful coping strategies (i.e., counseling and establishing a routine) and became more optimistic about the future. Students with disabilities looked back on their initial pandemic experiences and were able to take purposeful action to counteract the negative impacts that were brought about. By considering what was beneficial to their well-being, students with disabilities changed the way they coped to better manage their conditions or associated diagnoses. A female undergraduate student with anxiety and depression felt she could be more open when needing to ask for help, stating, *"I'm relying more on other people. Especially with the therapist, it's caused me to actually be more honest about what I'm feeling or when I need help."* A female undergraduate student with anxiety, depression, and bipolar disorder voiced the changes she experienced both as a person and in reference to coping when she stated in her spring interview:

> *"I kind of became a different person as the year went on and I think everyone has. I don't think anyone is the same person that they were when this whole thing started out, but as I grew and evolved and I learned more things, my coping strategies also kind of morphed along with it."*

At his spring interview, a male postdoctoral fellow with anxiety noted lifestyle changes, such as taking time for oneself, helped improve his coping abilities:

*"Now, I'm very much in better place in managing my mental health ( . . . ) think it has mostly to do with the looking at resources that are available. Overall, changing the lifestyle to have more of a routine that is not just work but also taking some time off from time to time."*

As mentioned by other students with disabilities, a male graduate student with anxiety felt his situation improved after prioritizing his mental health and reconnecting with others; he said:

*"I had the time to sort of focus on my mental health, which in the past kind of kept getting pushed aside because I was busy with work and school so it gave me the time to sort of reconnect with people in my life and start therapy, start medication for anxiety, I think in the beginning my mental health sort of deteriorated but as time has gone by, I think I've gotten better actually."*

### 3.2.7. Theme 7. "Support Systems"

The analysis revealed three sub-themes to illustrate who made up students with disabilities' support systems. These included: (1) peers, (2) family members, and (3) mentors.

*Peers*. Almost all students with disabilities noted their support system included friends, classmates, and/or significant others (e.g., boyfriend, girlfriend, or partner). A female postdoctoral fellow with asthma verbalized her support system during her fall interview when stating, *"My significant other is probably the big support and most of my friends who I would utilize as a support network are graduate students at UAB and other postdocs"*.

*Family*. Family members such as sisters, brothers, aunts, godparents, partners, husbands, and wives provided support to almost all students with disabilities. As mentioned by a female undergraduate with anxiety and depression, family was a big source of support in the fall interviews, when she voiced to the interviewer: *"I seek support from my family. We just kind of talk through things."* In the spring, many students with disabilities mentioned how members of their family, whether in-state or internationally, provided them support and comfort during the pandemic. For example, a male graduate student with anxiety discussed the support he felt from his family and boyfriend, saying, *"I live in a house with my brother and my boyfriend, so reaching out to them a lot more for help and talking to them about my mental health and my family as well have been very supportive"*.

*Mentors*. Many students with disabilities also mentioned mentors which included teachers, coaches, and other adults as being a part of their support system. For example, a female student with depression and anxiety illustrated in her fall interview how her supervisor or primary investigator (PI) was supportive during COVID-19 when she said: *"My PI is amazing and she's really good about prioritizing how we're doing our well-being above everything else"*.

### 3.2.8. Theme 8. "Perceived Support Seeking Barriers"

At the time of the initial fall interviews, four students with disabilities indicated that the question was not applicable because they always ask for support. Only two students in the spring indicated they always ask for help when needed, and thus, the question was not applicable. From the remaining students with disabilities, three themes representing why they do not ask for support were identified: (1) do not want to be a burden, (2) fear of judgment: self and public stigma, and (3) cultural barriers. These themes remained consistent from the fall to spring interviews.

*Don't want to be a burden*. When asked the reasons why students with disabilities did not seek help, even when help was needed, most students with disabilities voiced it was because they did not want to bother or burden others with their struggles. Many students with disabilities felt that by asking for help, it would inconvenience people or add to other people's problems. In the fall, a female undergraduate with depression, anxiety, and bipolar disorder voiced this when saying:

*"Sometimes I do get kind of sad and in a funk and I won't reach out to anyone because I feel like I'm a burden or they don't want to talk about that and it's just too pessimistic to talk about."*

During the fall interview, a female graduate student with depression and anxiety recently had surgery that impacted her mobility. She expressed her concern of being a burden to others and internalized her negative emotions:

*"I can't move when I want to. I can't do the things I want to and I really bottled it up inside and it's because being in a caring profession, the weight of putting so much on people, so sometimes I internalize a lot of things just so I won't be a burden but it just got too much to handle really. It was just too emotional for me."*

A male postdoctoral student with depression illuminated the difficulties gathering up the mental energy required to ask for help on top of being in a poor state of mind. While recognizing that others might feel the same, this student was reluctant to ask for help:

*"Sometimes I'm not in a frame of mind to talk to people about it, I don't have the energy to explain to people what's going on. With friends, everyone is going through something so the person you're reaching out to might not also be in good frame of mind and you can't just like unload your problems on that person. I think all the COVID-19 related breakdowns and anxiety played out like in those ways."*

*Fear of Judgment: Self and Public Stigma.* Many students with disabilities did not ask for help when needed because they felt that if they disclosed information regarding their struggles, they would be judged negatively. By not disclosing or seeking help, many chose to keep to oneself and engaged in withdrawal or disengaged coping. A female postdoctoral fellow with depression and ADHD became sick and did not want to disclose to her mentor. *"I didn't want to ask for her help because I didn't want her to think I couldn't do the job."* Another student, a female graduate student with MS and asthma, disliked asking for help because she felt she would then be defined by her disease; she voiced to the interviewer in the fall:

*"I don't want to let anybody down ( . . . ) I don't like to say I'm having a problem. I'm a whole person. I don't want to be defined by my disease, but that can be a problem and so I don't like to say I took on too much and that's why I'm not getting anything done and I'm exhausted and I feel like I failed."*

As was the case with fall interviews, many students with disabilities continued to illustrate they avoided seeking help because they did not want to disclose their challenges to others. For example, a female graduate student with anxiety felt embarrassed when expressing her mental health challenges during her spring interview, stating, *"Sometimes mental health struggles are embarrassing for me to talk."* In addition, in the spring, many students with disabilities expressed they did not have the mental energy to disclose their struggles to others, and thus did not. For example, a female graduate student with anxiety and depression indicated that having an anxiety disorder was already mentally taxing, and with the pandemic, no energy was left to seek help. She also felt that reaching out to others would cause others to label her as incompetent. She stated:

*"It's one of the things I constantly battle with. I'm having a lot of physical health issues but also if you have anxiety is it's very physically exhausting at times because you're constantly worried, especially to the point It's just debilitating, that gets me behind sometimes on work. I just can't give the energy. Then also having physical ailments ——I never want anybody to label me as a person like 'oh she's got so much stuff going on, she can't finish her work, she isn't meant to be in this program to do this'. I don't want to get labelled like someone lowered their expectations on me. It's been frustrating for myself because I want to give more, but I just really can't."*

*Cultural Barriers.* A few students with disabilities were reluctant to ask for help due to prior experiences of not receiving support when it was sought out. For example, a few students with disabilities did not ask for help because they grew up in a culture in which

asking for help was outside of their cultural norms. In other words, the stigma surrounding mental illness made seeking support from family and others difficult. A female graduate student with depression illuminated this when she told the interviewer in the fall about her family's lack of awareness or acceptance regarding mental health; she stated, "*I know that the people around me have a limited understanding of mental health, especially within my family. They've never really engaged in ( . . . ) admitting that you do have some type of issue with your mental health.*"

### 3.3. Unique Patterns across Racial Identity Groups and Grade Classification

Notable differences were seen regarding mental coping strategies, specifically for those engaging in positive or optimistic thinking as a form of coping. In the fall, when looking at White participants, only 1 of the 20 used positive thinking to cope. When looking at the remaining racial groups, all being persons of color, 31.25% (5/16) engaged in positive thinking to cope. However, participants who were biracial did not use positive thinking as a coping mechanism at all until the spring. In the fall, notably, when looking at undergraduate students (*n* = 14) and mental coping strategies, none used positive thinking or engaged in spirituality-based practices to cope. Conversely, within the 17 graduate students, 6 incorporated optimistic thinking to cope. For the five postdoctoral fellows, only one used positive thinking to cope across both interviews. By the spring, only one undergraduate student partook in spiritual practices, and only two engaged in thinking positively to cope. Graduate students were the only grade cohort that engaged in helping others as a relational coping mechanism (4/17) in both the fall and spring interviews.

### 4. Discussion

The findings from this study illustrate the distinct experiences pertaining to coping and help-seeking behaviors for college students and postdoctoral fellows with disabilities and/or pre-existing conditions during the COVID-19 pandemic (fall 2020–spring 2021). To cope with the drastic changes and unprecedented challenges resulting from the COVID-19 pandemic, students with disabilities employed a variety of behavioral, mental, and relational coping strategies. Enjoying hobbies, spending time outside, and establishing a routine were the most endorsed of all the behavioral coping strategies across the fall and spring. For mental coping strategies, counseling was utilized the most by students with disabilities, followed by distraction. Counseling was the primary mental coping strategy in the spring as well; the second most used was positive thinking or optimism. Most students with disabilities emphasized the importance of keeping in contact with others (i.e., relational coping).

Whether coping in response to a specific event or their perception of how they have coped overall, most students with disabilities felt satisfied with their coping when they were able to manage and reduce the stressors they were experiencing. This was consistent into the spring. In the fall, students with disabilities noted that a reduction in their levels of isolation was the second most endorsed determinant of successful coping. However, by the spring, this changed, as more students with disabilities began reframing how they viewed the pandemic, adapting to their struggles and utilizing a positive mindset. Despite the additional stress induced by the COVID-19 pandemic, over time, students with disabilities were able to tap into their resiliency and adapt to cope more effectively. As students with disabilities utilized helpful and adaptive coping strategies (i.e., counseling, setting a routine, and staying connected to others), they became more optimistic about their future and felt more confident in their abilities to cope. In line with the TMSC moderating construct of reappraisal [21], students with disabilities looked back on their initial pandemic experiences and were able to take purposeful action to counteract some of the negative impacts that were brought about by the pandemic. By considering what was beneficial to their well-being, many students with disabilities changed the way they coped to better manage their conditions or associated diagnoses.

These findings speak to the untapped resiliency and perseverance present in the disability community within higher education, an area of research that is unsurprisingly rather limited. A scoping review conducted by Murphy and Markey [40] on the impact of COVID-19 on people with pre-existing conditions identified studies reflecting resilience in the disability community from a global standpoint [25,41,42]. Resiliency is "the process of, capacity for, or outcome of successful adaptation despite challenging or threatening circumstances" [43] (p. 426). In times of chronic stress, such as the pandemic, developing a sense of resilience is crucial. Furthermore, studies on resiliency provide plausible explanations for why students with disabilities in this study were able to persevere and adapt versus those who did not. Research shows acceptance of change, spirituality, having a sense of community, and quality social support are constructs that foster resiliency [24,44,45]—all themes shown in this study. Conversely, burnout, the presence of negative emotions, low self-efficacy, and lack of support are barriers to resiliency and can negatively impact academic success and adjustment [44,46]. Thus, it appears the components of reappraisal, resilience, and utilization of social support hold the potential to mitigate COVID-19 impacts. As such, these components should be considered when developing interventions or resources to support the success of students with disabilities as they continue their educational journey.

Although students with disabilities coped with multiple strategies (i.e., behavioral, relational, and mental), stigmatizing barriers kept many of these students from obtaining resources, fearing that help seeking would result in negative judgment or devaluation from others. This judgment was reflected internally as well, inciting feelings of inadequacy and embarrassment. Alongside mental and physical exhaustion that can stem from living with a pre-existing health condition or disability, some students with disabilities isolated themselves instead of asking for help, which ultimately was more hurtful than helpful. Lastly, most students with disabilities did not reach out for help during times when help was needed because they did not want to be viewed as burdensome or add to others' own struggles. Reluctancy to disclose and seek help can stem from wanting to avoid the deficit-focused view associated with disability (e.g., less than, weak, lazy, etc.) [47]. Further supported in the present research, most students with disabilities feared that upon disclosure, they would be met with negative responses, labels of inadequacy, and devaluation of their perceived ability. According to our interviews, those views were mirrored internally as well, leading to feelings of inadequacy and embarrassment. In line with previous research on self-stigma (i.e., incorporating negative stereotypes into self-view), how we view ourselves can be highly influenced by how others view us (e.g., professors and peers) [48–50]. In a similar way that disability is socially constructed, researchers posit that self-concept is also socially constructed; thus, our finding that students with disabilities did not reach out for help due to fears of being judged negatively or burdensome to others are examples of internalized or self-stigma [48,51,52]. A reduction in self-esteem, self-worth, empowerment, and community participation in addition to hindering social support and exacerbating symptomology are just some of the negative effects of internalized stigma [50,53,54]. Social or public stigma (i.e., prejudice and discrimination towards disabilities), a precursor to internalized stigma, is also related to an increased sense of isolation [51,55], which, as shown in this research, was a dominating stressor for many college students with disabilities.

Thus, if higher education does not address this inherent ableism clearly influencing students' self-view, a vicious cycle of devaluation, fear of judgment, shame, isolation, and reluctance towards support is likely to continue and widen as more students begin to navigate life with a disability. One factor impacting an individual's likelihood of self-disclosure is related to their self-advocacy skills, or the ability to communicate one's needs and obtain resources to meet those needs [56]. In other words, the stronger one's self-advocacy skills are, the more likely they will self-disclose. Furthermore, one is more likely to have stronger self-advocacy skills when they have a more positive sense of self pertaining to their disability and a strong sense of belonging [12,13,57]. As it relates to education, having a positive self-concept and strong self-advocacy skills are large predictors for academic success [58] Considering that this study illustrates an absence of positive

identity acceptance related to disability, on top of the significance of the developmental period which the majority of the sample is in, there is still much to be done.

*Use of the TMSC Framework to Interpret Findings*

The themes revealed from the analysis intersect with the tenets of the TMSC (Figure 1). As previously noted, this model evaluates responses to stressful life events (i.e., the COVID-19 pandemic) and the processes of how one copes in response to these stressors [20]. By triangulating the themes within this model, we can better understand the relationship between perceived stressors, coping mechanisms, and various psychological outcomes. In addition, by mapping overlapping themes with the tenants of the TMSC, we can further elaborate and validate this study's findings. Based on the TMSC, the COVID-19 pandemic is the *stressor*. When faced with a *stressor* or difficult situation, (in this case, the COVID-19 pandemic), students with disabilities evaluated the pandemic (*primary appraisal*) as highly distressing both to their mental health and their academic performance and had differing evaluations regarding their ability to cope (*secondary appraisal*). Within the construct of primary appraisal is *perceived susceptibility and severity*, which pertains to COVID-19 in this case. Individuals with disabilities are at a higher risk for getting COVID-19 and potentially dying from COVID-19 than their non-disabled peers [59,60].

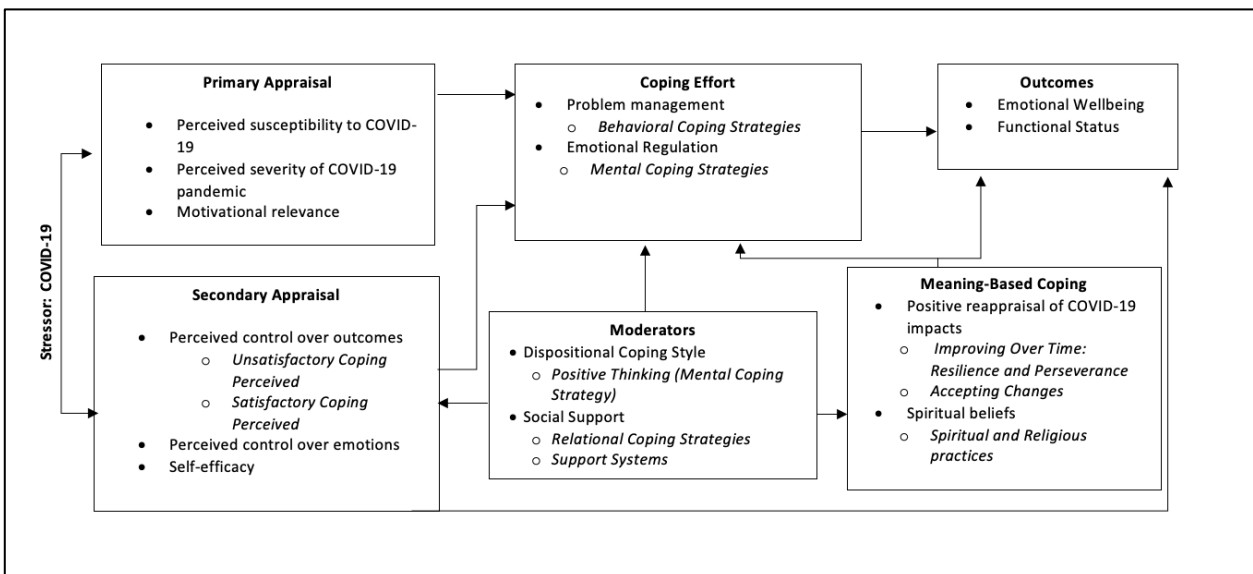

**Figure 1.** Impact of the COVID-19 pandemic on college students and postdoctoral fellows with disabilities or pre-existing conditions outlined by the Transactional Model of Stress and Coping, modeled after Lazarus, Cohen, Folkman, and Moscowitz [20,21,61].

Once a stressor is perceived to be a threat, *secondary appraisal* takes place, in which one assesses their perceived ability to change or control their emotions and the outcome of the situation [20]. Individuals assess the availability of their coping resources and reflect on their ability (self-efficacy) to successfully cope in response to the stressor of COVID-19. As such, *unsatisfactory and satisfactory* coping themes fit within the model's construct of secondary appraisal. Students with disabilities struggled more in the fall than in the spring in terms of their perceived ability to cope and manage through the pandemic. When coping was deemed "*unsatisfied*", many students also felt unsatisfied with themselves as they engaged in *negative habits, had difficulty maintaining pre-pandemic coping abilities, and were unable to regulate their negative emotions*. Research shows when a stressful event is perceived as unmanageable and/or overwhelming, and the individual does not have the resources to adequately address the stress, more stress ensues. Although some amount of stress can be adaptive, this negative form of stress leads to negative psychological *outcomes* [20,22]. In the fall, we saw some students with disabilities expressing high levels of stress, noting the

symptoms associated with their condition (i.e., depressive episodes, difficulty focusing, and relapses in condition) had worsened. For example, due to additional stressors brought about by the COVID-19 pandemic, a female undergraduate student with anxiety and depression noted her mental health was detrimentally affected. Many students with disabilities noted that because of the exacerbated symptomology they were experiencing, their coping mechanisms were not as effective as they had been prior to COVID-19. However, as the year continued into the spring 2021 semester, students with disabilities felt their coping abilities improved as they were able to better manage their stress, which they felt resulted in an improvement to their well-being.

The coping strategy themes (i.e., behavioral, mental, and relational) aligned with the TMSC dimensions of coping: *problem-focused coping* and *emotion-focused (or regulation) coping*. Problem-focused coping is action-oriented and involves utilizing physical strategies to change the situation or problem at hand [20,22]. For students with disabilities, the stress amplified from the circumstances of the pandemic was dealt with using a variety of both problem-focused and emotional-focused coping mechanisms. For example, *behavioral coping strategies* such as *establishing a routine*, engaging in *outdoor physical activity*, and partaking in *hobbies* helped students with disabilities *manage their stress, feel less isolated*, and helped maintain their pre-pandemic levels of *productivity*. These techniques were endorsed more often than other behavioral strategies (i.e., *decreased media exposure, substance use, and poor eating habits*) from the fall to spring interviews. These problem-focused coping strategies are deemed most adaptive in situations where the stressor is controllable [22]. The COVID-19 pandemic felt far from controllable to many students with disabilities, as they expressed feelings of uncertainty and fear regarding what the future might hold (i.e., *instability stress*). Feelings of instability, uncertainty, and anxiety are all heightened during emerging adulthood compared to other developmental periods [23,62], and most likely exceeded the level of stress that is adaptive as the pandemic exacerbated these emotions [14,63]. When a situation or stressor is deemed unchangeable, the TMSC model states that emotion-focused coping is most adaptive. Emotion-focused coping is even more effective when used simultaneously with problem-focused coping [20,22]. Consistent with prior COVID-19 research, students with disabilities engaged in both forms; however, the available studies are limited. For example, Kourea and Christodoulidou [64] found undergraduates with disabilities were able to reduce their stress and anxiety by establishing or re-establishing a sense of routine. In this study, the results show routine was used by participants across all education levels; however, undergraduate students utilized this strategy the least compared to graduate students and postdoctoral fellows. It may be that since graduate students and postdoctoral fellows have been in academia longer than undergraduates, they have more familiarity with establishing structured routines.

As such, future studies are needed to investigate the reasoning behind why these differences within grade classification are present. For mental or *emotion-focused* coping strategies, counseling was the most endorsed coping strategy by students with disabilities during both the fall and spring semesters. The second most endorsed strategy was *distraction* in the fall, and for the spring, it was using *positive thinking or optimism*. However, differences within grade classification were present. Specifically, undergraduates, compared to all other grade classifications (i.e., graduate students and postdoctoral fellows), did not engage in positive thinking as a mental coping strategy during the fall, and only one noted this coping strategy in the spring.

Most students with disabilities emphasized the importance of keeping in contact with others (i.e., *relational coping strategies*) and discussed how the pandemic increased their efforts to seek out virtual opportunities for social interaction. As previously noted, social support and optimism can foster a sense of resiliency and moderate the stressor's effect on psychological outcomes [24,45,65]. In line with the TMSC, *social support, positive thinking, and spirituality* can moderate the effects of stress [61,66], which were all themes found in this study. Thus, universities should develop opportunities for students to connect and provide workshops to increase positive thinking skills. As indicated in the interviews,

students with disabilities' support systems involved *peers, family, and mentors*; however, with border restrictions, international students participating in the study were not able to visit their family or return home. Accordingly, attention should be given to these students as they might need additional social support. Overall, building on the TMSC [20], this study illustrated that mental coping strategies (positive thinking/optimism), relational coping strategies (staying connected with others), and the presence of social support (counseling, family, and peers) were important factors for the successful management of stress for students with disabilities during the COVID-19 pandemic.

Despite the significant contribution to the literature this study provides on college students with disabilities during the COVID-19 pandemic, this study also has some limitations. First, attrition was a problem, as it is in most studies. Fortunately, it did not impact the study population in terms of gender or educational level. However, it did in terms of race, as four of the participants who did not complete their second interview were African American. It is important to note that the results speak to a specific collegiate population at a specific university, and thus, one should be cautious in generalizing results to the community of students with disabilities. Second, the study population did not have a diverse range of disabilities. Future studies should seek to recruit a more diverse range of disabilities or conditions to add to the present literature on students with disabilities' experience during the COVID-19 pandemic. In part, this is since the parent study was not specifically designed to investigate students with disabilities. It is important to note that the sample would most likely have identified more students with disabilities if additional context was included in the interview questions, since students may not directly identify with a disability, feel comfortable discussing it, or fully understand what a disability is. Additionally, given the stigma associated with having a disability, students avoid disclosure and participation in disability-focused studies [47,67–69].

## 5. Conclusions

This study provided detailed information on how college students and postdoctoral fellows with disabilities or pre-existing conditions are coping with the COVID-19 pandemic. Additionally, this study illustrated underlying barriers college students and postdoctoral fellows with disabilities or pre-existing conditions are facing towards seeking support during the COVID-19 pandemic. Our findings suggest students are less likely to seek help because of self or public stigma pertaining to mental health and disabilities. They expressed that upon asking for help, they feared they would be met with negative judgment and thus did not seek help. Importantly, this study highlights the resiliency and strength of students with disabilities, and thus, universities should strive to bolster factors that promote resiliency (i.e., social support, positive thinking, and self-advocacy). Collectively, these findings highly emphasize the need for the conversation to change. There can be copious amounts of resources provided and offered to students, but if society does not address (1) the way that disability is perceived and taught and (2) the deep-rooted ableism that plagues this country, disability and mental health resources will continue to not be sought out, and inequitable gaps are likely to continue widening. We can provide a multitude of resources and support to address student's needs, but if we live in a society where we feel too ashamed or embarrassed to ask for support and accept that support, such efforts will go wasted. Whether in research, policy, or other public health domains, the voices of historically marginalized communities are critical to be heard and to produce real positive change. In addition, we must forgo ableist assumptions that have long guided the dialogue of higher education. This study's results suggest that the TMSC could be helpful in conceptualizing the impact of COVID-19 and processes related to stress on psychological outcomes and avenues to consider in future intervention designs for mitigating the impact of stress for students with disabilities. Further research should focus on developing interventions that strengthen the dissemination of mental health awareness and address stigma related to mental health and disability present in higher education. Looking to a future post-COVID-19, we have a responsibility to protect, empower, and

involve students with disabilities in the hopes of creating a more just and equitable space for all. Echoing the stance of the disability rights movement, this work emphasizes the importance of "*Nothing about us, without us*".

**Author Contributions:** Conceptualization, C.W., R.G.L., A.S. and L.S.; methodology, C.W., C.O., R.G.L., A.S. and L.S.; formal analysis, C.W., C.O. and R.G.L.; resources, R.G.L., A.S. and L.S.; writing—original draft preparation, C.W.; writing—review and editing, R.G.L. and K.U.; visualization, C.W.; supervision, R.G.L.; project administration, R.G.L., A.S. and L.S.; funding acquisition, R.G.L., A.S. and L.S. All authors have read and agreed to the published version of the manuscript.

**Funding:** This research was funded by the UAB 2020 School of Public Health Back of the Envelope Award (PI: Robin Gaines Lanzi, Lisa Schwiebert, and Angela Stowe) and supplemental internal funds from the Department of Health Behavior (Robin Gaines Lanzi), the Department of Cell, Developmental, the Integrative Biology (Lisa Schwiebert), and the Office of Service Learning and Undergraduate Research, all at the University of Alabama at Birmingham.

**Institutional Review Board Statement:** The study was conducted in accordance with the Declaration of Helsinki and approved by the Institutional Review Board of The University of Alabama at Birmingham (UAB) (IRB-300005302 and 5 June 2020).

**Informed Consent Statement:** Informed consent was obtained from all subjects involved in the study.

**Data Availability Statement:** The data presented in this study are protected by the Institutional Review Board protocol. Data requests should be submitted to the senior author, MPI of the study (rlanzi@uab.edu). The request will be reviewed by the MPIs of the study in accordance with the Institutional Review Board policies and procedures for the protection of human subjects in research.

**Acknowledgments:** We would like to thank the undergraduate students, graduate students, and postdoctoral fellows who participated in the study. We would also like to thank the interviewers who helped conduct the qualitative interviews.

**Conflicts of Interest:** The authors declare no conflict of interest.

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
