# Peer review of "Coping Strategies and Help-Seeking Behaviors of College Students and Postdoctoral Fellows with Disabilities or Pre-Existing Conditions during COVID-19"

_disabilities, doi:10.3390/disabilities3010006_

Round 1

Reviewer 1 Report

I commend you on a very interesting paper, addressing the experiences of disabled students in the context of Covid-19. There has been a notable gap in evidence exploring experiences among disabled students, and I believe this paper makes a valuable contribution. Overall, I found the paper to be well-written, clearly organised, rigorous in its analysis and strong in terms of its contribution. I am very happy to recommend this paper for publication, but I would like to identify a few changes which could enhance the paper.

1.       There are language, syntax and grammatical errors throughout the paper. Some examples are found on lines 46, 85, 245, 326, 438, 441, 544, 621, 630, 656, 757, 769, 790, 811, 911, 916. Careful editing throughout is necessary.

2.       I am unable to find information on how the sample was recruited and selected, this is an important omission which should be addressed. I gather the data stems from a larger study. However, it is important that this paper describes all steps, including sample recruitment, clearly. The sample is unbalanced, in many respects, but it is not clear how this arose. I don’t necessarily consider the sample profile being a limitation, but the implications should be discussed.

3.       I am not sure it is necessary to include the identifiers, such as a male postdoctoral fellow with anxiety. I wonder would it be sufficient to include male postdoctoral fellow. Given the scale of this study, I don’t think comparison across disability types/conditions is feasible, rather I think you are aiming to elicit views and experiences across a diversity of students.

4.       The paper uses sweeping statements throughout, particularly regarding ‘students with disabilities’. I would strongly urge caution here. Given the methodology employed and the nature of the study, the results don’t really speak to all students with disabilities (even within your sample), but rather to some students or many of the students we engaged with …..

5.       In order to improve the information presented on the methodology (and sample selection), I suggest reducing some of the results material. The paper is very long, and I think repetitive in places.

Author Response

Hello! Thank you very much for your comments and inquiries. Thank you so much for your patience, understanding, and valuable time. I am very grateful. Please see attachment.

Reviewer 2 Report

Minor edits: line 46. would read better with "and thus" rather than "as thus"; line 68, wellbeing should be hyphenated; line 85 remove "to"; line 244: aren't exercise and workouts the same thing? Is that redundant? Line 275 reads awkwardly, "A female graduate student with ADHD and anxiety voiced in her spring 274 interview the multiple coping strategies she engaged in, one being having a set schedule..." Suggestion, reword to say, "A female graduate student with ADHD and anxiety voiced in her spring 274 interview the multiple coping strategies she engaged in. For example,  having a set schedule...."; line 305 "eat" should be "eating"; line 306 in the quote, was the ' in "day's" in the original quote or is that a typo? line 309 needs a comma after "In the spring"; line 326, was "of" in the original quote or is that a typo? Line 367 would read better with a hyphen in mental based; line 418, did "If" have a capital in the original? If not, it should be lowercase. line 437: shouldn't "productive" be "productivity"? Line 438: "the" should be "they"; Line 441: remove "and"; line 467 remove the first "was"; line 512 should be "led" instead of "lead"; line 544: after "desire" add the word "to"; line 621, remove the word "Analysis" at the beginning of the sentence; line 625: remove "and'; line 630 should read "they coped", not "the coped"; line 656, 'disabilities' needs an apostrophe as it's possessive in this case; line 718, need "for" before "help"; line 757, "families lack awareness" should read "family's lack of awareness"; line 769, "form to cope" suggest "form of coping" to read more smoothly; thi sentence, starting on line 770 needs reworked, "When 770 looking at remaining racial groups, all being POC, this percentage was higher as 31.25% 771 (5/16) of these participants used positive thinking to cope." line 780 -"heling" is that supposed to be "helping"? Line 792: "strategies" should be strategy - the noun is counseling, which is singular. Line 792, after "as well" should have a semicolon. Line 811, last word should be "they" not the; Line 818 has "adaptation" but previously "adaption" was used. May want to consider being consistent and using the same word. Line 911: "unable regulate their negative emotions" - Needs "to" - "unable to regulate...." Lines 911 - 914: "Research shows when a perceived as unmanageable and/or overwhelming, and the individual does not have the resources  to adequately address the stress, more stress ensues and although some amount of stress can be adaptive, this negative form of stress leads to negative psychological outcomes [6, 9]." This sentence is confusing as it is missing a piece at the beginning - "Research shows when a perceived..." What? Perceived what? Add that and the sentence should make sense. Lines 915 & 916: "In the fall, we saw students with disabilities expressing high levels of stress, noting the symptoms associated their condition (i.e., depressive episodes, difficulty focusing, 916 relapses in condition) had worsened." Need 'with" after "associated". Line 923 into 924 has the word "well-being" with a hyphen. This might be due to the line break, if it's due to spelling the word that way, that is correct. Change all other uses of the word to be hyphenated. Line 929: "with in" should be one word. 

This paper is quite informative to the field. I have no corrective feedback on content or outline, only grammar, spelling, and mechanics as noted above. It is highly likely that I missed listing some of the errors. A careful editing will help improve the overall clarity of the paper.

Author Response

(The authors gave the same response as above.)

Reviewer 3 Report

Dear author(s),

Thank you so much for the opportunity to review the manuscript “Coping Strategies and Help Seeking Behaviors of College Students and Postdoctoral Fellows with Disabilities or Pre-Existing Conditions during COVID-19.” As you have mentioned in the manuscript the COVID-19 pandemic had and continues to have a global and multifaceted impact on public health, especially on the marginalized and vulnerable populations. Your topic is well timed with events that affect the global society, and it is important that we also explore how college students with disabilities behaved during the pandemic. The study can contribute to the literature by helping higher education institutions better prepare for similar events in the future.

Overall, the manuscript is well written however, some major revisions must be made in order to comply with the requirements for high quality of research. The manuscript can be improved by addressing the requirements listed in the literature for research guidelines for higher education and disabilities (see https://files.eric.ed.gov/fulltext/EJ1293017.pdf ). I am listing below my comments.

1.       Introduction: I would suggest the following:

a.       to expand and improve this first section by discussing in more depth how students with disabilities were impacted by the pandemic, in particular those attending higher education institutions.

b.       At page 2, line 80, you have mentioned that coping strategies for higher education students were not researched. Was this topic researched for any category of students with disabilities such as, for example, the secondary level? There is a tremendous amount of research that focused on students with disabilities during the pandemic. Just at the first glance, doing a Google search, I have found some research articles that were not cited in your manuscript.

c.       Page 3, first paragraph: change the past tense and state the research question

2.       Methods:

a.       How were the participants selected?

b.       Did all the participants attended the same university?

c.       How long was each interview? Who conducted the interview?

d.       Please describe the development of data-collection instrument

e.       Please clarify the following sentence: “Once interviews were completed, questions focused on probing how students are coping, how they feel about how they have been coping, who is their support system, and in times where support was needed, what prevented them from seeking help.” Pg. 3, line 114-116.

f.        Please clarify the following sentence: “All adaptions or changes were discussed with the team’s primary qualitative coder (C.O) under the direction of the Principal Investigator to ensure continuous validity.”(Page 4, line 161-163). Describe the team.

g.       Provide more details about the inter-coder agreement. What was the %? Who were the coder(s)? The first author was the only one mentioned to analyze the data.

3.       Results:

a.       Did the IRB and the participants allowed the author to state the name of the University? Page 5, line 184

b.       How did the author(s) collect the demographic characteristics data?

4.       Discussion:

a.       Any research on how students with disabilities behaved during the pandemic? Did students with disabilities behaved different than those without disabilities?

5.       Add limitations of the study.

Author Response

(The authors gave the same response as above.)

Round 2

Reviewer 3 Report

Thank you for addressing all my comments!

Author Response

  1. Abstract to better understand is a split infinitive

Thank you, this comment is appreciated. To address this, I removed ‘better’ on line 19, as noted in highlight and with the use of the strike through function.

  1. Line 71 page 2. It appears there are five components. Number 4 is repeated twice.

Thank you for pointing this out. On line 70, I have removed “4) coping style” from the list, it was not supposed an individual component because coping styles are encapsulated in the secondary appraisal component as moderating factors. However, there are 5 components, which I mistakenly did not include the 4th in the list. On line 71, this component “meaning-based coping” was added. This is further discussed on line 91.

  1. Line 117, page 3. What was the study methodology? Should cite a named methodology and justify that choice.

Thank you for your inquiry, it is appreciated. I agree that a citation and justification for the methodology should be included in this section. Thus, I have added in the following section directly under the Methods starting at line 135 stating:

2.1. Design.

For this study’s purposes, a phenomenological approach was best fit as it seeks to understand a group’s lived experiences of a phenomenon, describing the essence of what is being experienced and how it is being experienced.The central phenomenon for this study is the self-described coping strategies and help-seeking behaviors of college students and post-doctoral fellows with disabilities or pre-existing conditions during the time of the COVID-19 pandemic. When taking a phenomenological approach, data collection methods primarily involve in-depth interviews, as they incorporate open-ended questions to elicit rich information and detailed perspectives on topics exploratory in nature. In addition, given the exploratory nature of the study and the complexities of the COVID-19 pandemic, using a phenomenological qualitative research design was warranted. Institutional research ethics (IRB) approval for this project was obtained from the University of Alabama at Birmingham (UAB).

Subsequently, section numbers were updated to reflect the addition of the section.

  1. Ibid. By triangulating does that mean the TMSC was used deductively?

I appreciate this inquiry. Yes, the emergent themes were found to map most closely to the TMSC Framework allowing for validation and expansion of this model. Triangulating emergent themes and patterns with the TMSC provided a useful framework for understanding and expanding on the qualitative analysis and validating emergent findings.

  1. Line 134, page 3, What is meant by “The longer 8-month study.
    Thank you for your inquiry, this study utilized secondary data from a larger, mixed methods, longitudinal study conducted at the University of Alabama at Birmingham (UAB) titled COVID-19, Race, and Student (undergraduates, graduates/professional) and Postdoctoral Fellow Mental Health Study (PIs: Drs. Robin Lanzi, Angela Stowe, and Lisa Schwiebert). The larger study lasted 8 months. To add more context to that phrase, I restructured the first part, on line 149, stating: “The parent study, in which this study utilized secondary data, was an 8-month mixed methods longitudinal study conducted at UAB. It included a baseline survey.... “

  2. Line 157, page 4. Braun and Clarke describe their approach now as reflexive thematic analysis.

Thank you for this enlightening comment. After your comment, following a re-review of Braun and Clarke’s recent work, I am now aware of this terminology change. Definitions of thematic analysis as well as methodological guidance for its use have been described in multiple, and varied ways, thus leading to considerable confusion among researchers (Braun & Clarke, 2019; Terry et al., 2017). I would explain the approach as a "hybrid" approach where you begin with a deductively produced codebook and add to it, inductively, as you gain new insights into the data.

Reflexive was added on line 175, 195, 197-198, 214, 219-220, 237, 258

169, 191, 193, 218, 240, 250, 604

Additionally, citations were updated and added to address this change.

  1. Line 182 page 4. Braun and Clarke would probably call this a code book approach rather than their reflexive thematic analysis.

I would agree that this aspect is more in line with a codebook approach. However, as mentioned in the response above, we used more of a hybrid approach as defined by Braun and Clarke. I want to make sure to acknowledge that because a codebook was used for the primary parent study, it was used as a guide to reflect on themes for this paper. Given the use of secondary data, and the flexibility of reflective thematic analysis, this study is an example of utilization of a codebook as part of the reflective thematic analysis. To further justify, I have provided excerpts from two papers, illustrating how these can be used:

  1. Braun and Clark, 2020

Codebook approaches (e.g. King & Brooks, 2018; Ritchie & Spencer, 1994) Combine the qualitative research values of reflexive TA (Braun and Clarke, 2020).

  1. Ayre, Julie & Mccaffery, Kirsten. (2021). Research Note: Thematic analysis in qualitative research. Journal of Physiotherapy. 68. 10.1016/j.jphys.2021.11.002.

Codebook thematic analysis lies between codebook reliability and reflexive thematic analysis and is recommended for applied health research. 13  In this approach, themes are generated and charted into a framework; however, this framework does not determine reliability or accuracy, as it would for content analysis. Rather, the framework exists to help further develop themes, and is particularly useful for integrating efforts from multiple researchers.  14  Whilst the output of the analysis is ‘themes’, these can range from topic/domain summaries through to more fully developed themes. Similar to reflexive thematic analysis, codebook thematic analysis affords flexibility in the extent that inductive and deductive coding are used. Given implementation science's emphasis on theoretical frameworks, these studies often take a mostly deductive approach (themes align with components of these frameworks), whilst remaining open to other themes, which are developed through an inductive approach.  15 

  1. Line 221, page 5. Dissertation? First time this term is used.

Thank you for your inquiry. This study was a part of my dissertation and thus was originally written that way. Thank you for catching this accidental mistake. As such, I have switched out the word “dissertation” with “study” on Line 236, as it is more appropriate in this context.

  1. Line 879, page 19. The term obsolete is not correct

I appreciate and agree with your comment. I believe “limited” (line 905) is much more appropriate in this case given the vast gap between the number of studies on students during COVID versus studies focused solely on students with disabilities. Although research is emerging in this much needed area, there is still more information needed to best understand these students.

  1. Line 979-987, page 22. Seems like new data.
    Thank you for your comment. If in reference to the study, it seems the quote/data that supports the example regarding the participant with epilepsy was mistakenly omitted. I believe it would be appropriate to remove that example then (Lines 1010-1012, marked in yellow highlighter with the strike through function) However, the finding regarding exacerbated symptomology effecting coping stems from within the “Unsatisfactory Coping Perceived” theme. To address this comment, I have interjected the following to further support the data on the lines. Starting at line 1007-1009: “For example, due to an inability to additional stressors brought about by the COVID-19 pandemic, a female undergraduate student with anxiety and depression noted her mental health was detrimentally affected.”

  2. Line 1013, page 22. Would replace “all grade classifications” with a different term… ? across all education levels.

Thank you for your comment, I agree and believe the original sentence structure could be better. Thus, at line 1041, “Although our study showed that all grade classifications engaged in routines to cope” was adapted to “In this study, results show routine was used by participants across all education levels; however, undergraduate ...”.

During the final readthrough, the following editing were addressed:

  • Line 78 – there was an incomplete sentence. To fix this “In other words, an individual’s perception on their ability to change the stressful situation at hand as well as regulate associated emotions to cope successfully.” was changed to “An individual’s perception of their ability to change the stressful situation at hand as well as one's perceived ability to regulate associated emotions are examples of secondary appraisal.”
  • Line 121 – the following sentence was added “Additionally, findings identified student’s perceived barriers from seeking out resources or help when needed.”
  • Line 220, the word “sub themes” was missing a dash, thus it is now written out “sub-themes”
  • Line 605 was mistakenly italicized (Of the students who were dissatisfied with their coping strategies)
  • Line 1047 grammatical fix of TSMC to TMSC
  • Incorporated “Acknowledgements” section following IRB section at the end of the paper.

Of note, the figure in my paper incorporates components from the initial paper I submitted for this journal (Mental Health Experiences of College Students and Postdoctoral Fellows with Disabilities or Pre-Existing Conditions during the COVID-19 Pandemic: A Qualitative Study). In the instance that the first paper is rejected, given its decision is pending, I want to provide an adapted figure that does not incorporate the components of the initial paper (inserted on following page). In addition, if rejected the following statements in this paper would be omitted to remove confusion (lines 998): “Within the construct of primary appraisal is perceived susceptibility and severity, which pertains to COVID-19 in this case, and is reflected in participant statements within the themes illustrating students perceived mental health impacts resulting from COVID-19. Most relevant, infection stress exemplified that many students with disabilities were extremely worried about getting COVID-19 given that their condition puts them at a higher risk for getting COVID-19 and potentially dying from COVID-19 [51, 52].”

Figure 1. Impact of the COVID-19 pandemic on college students and postdoctoral fellows with disabilities or pre-existing conditions outlined by the Transactional Model of Stress and Coping, modeled after Lazarus, Cohen, Folkman, and Moscowitz [19, 20, 53].
